# Emergent quasiparticles at Luttinger surfaces

Michele Fabrizio [1✉]

In periodic systems of interacting electrons, Fermi and Luttinger surfaces refer to the locations within the Brillouin zone of poles and zeros, respectively, of the single-particle Green's function at zero energy and temperature. Such difference in analytic properties underlies the emergence of well-defined quasiparticles close to a Fermi surface, in contrast to their supposed non-existence close to a Luttinger surface, where the single-particle density-of-states vanishes at zero energy. We here show that, contrary to such common belief, dispersive 'quasiparticles' with infinite lifetime do exist also close to a pseudo-gapped Luttinger surface. Thermodynamic and dynamic properties of such 'quasiparticles' are just those of conventional ones. For instance, they yield well-defined quantum oscillations in Luttinger surface and linear-in-temperature specific heat, which is striking given the vanishing density of states of physical electrons, but actually not uncommon in strongly correlated materials.

[1] International School for Advanced Studies (SISSA), Via Bonomea 265, 34136 Trieste, Italy. ✉email: fabrizio@sissa.it

Periodic systems of interacting electrons are classified as Landau's Fermi liquids[1,2] whenever they possess coherent 'quasiparticles', namely single-particle excitations with dispersion $\epsilon_{qp}(\mathbf{k})$ in momentum space whose on-shell lifetime at zero temperature, $T = 0$, diverges on the surface within the Brillouin zone where $\epsilon_{qp}(\mathbf{k})$ is equal to the chemical potential, which we hereafter take as the zero of energy. Those quasiparticles therefore contribute, e.g. to thermodynamic properties essentially like free fermions, thus with a specific heat linear in $T$ and a finite paramagnetic susceptibility.

In the classical microscopic derivation[3,4] of Landau's Fermi liquid theory, the quasiparticles correspond to poles of the $T = 0$ Green's function at $\epsilon = \epsilon_{qp}(\mathbf{k})$ and $\mathbf{k} \to \mathbf{k}_F$, where $\mathbf{k}_F$: $\epsilon_{qp}(\mathbf{k}_F) = 0$ defines, in this case, the Fermi surface. That conventional quasiparticle thus shows up as a peak in the single-particle density of states (DOS) that, as $\mathbf{k} \to \mathbf{k}_F$, moves towards $\epsilon = 0$ and, simultaneously, grows and narrows.

Therefore, the absence of a peak in the single-particle spectrum measured, for instance, in angle-resolved photoemission spectroscopy (ARPES), is commonly interpreted as the non-existence of quasiparticles, all the more when the peak is replaced by a pseudogap like in underdoped cuprates[5–7], and attributed to a breakdown of Landau's Fermi liquid theory. However, it may happen that, despite such anomalous spectral features, other physical quantities behave as in conventional Fermi liquids. This is, for instance, the case of quantum oscillations, usually interpreted as characteristic signatures of well-defined quasiparticles with their own Fermi surface[8], which are observed also in the pseudogap phase of cuprates[9–12] together with a standard linear behaviour of the electronic specific heat[13–15]. Such Janus-faced character of some correlated materials becomes amazing, almost paradoxical, in $SmB_6$, an insulator that shows quantum oscillations characteristic of a large Fermi surface[16], and in spin liquid Mott insulators, which seem to display conventional Fermi liquid-like magnetic and thermal properties[17–20].

That anomalous physical behaviour suggests the existence of new quantum states of matter[21–26], but in some cases, it might simply indicate that Landau's paradigm of Fermi liquids applies to a much broader class of interacting electron systems than conventionally believed[27,28], even those where perturbation theory breaks down and thus Landau's adiabatic hypothesis[1,2] is not valid.

Perturbation theory can break down already at weak coupling because of Fermi surface instabilities, as in one-dimensional interacting Fermi gases[29], or only above a critical interaction strength. In the latter case, the breakdown is commonly associated with the emergence at $T = 0$ of a zero-energy pole in the self-energy, thus a zero in the Green's function. Such singularity may either signal the transition to a Mott insulator, as observed, e.g. within dynamical mean field theory[30], or, if the system remains compressible, the birth of a Luttinger surface[31] where the Green's function develops zeros at $\epsilon = 0$, in place of the former Fermi surface, on which the Green's function has instead poles. Since a zero of the $\epsilon = T = 0$ Green's function corresponds to a pseudo- or hard-gap, this circumstance looks at odds with the common definition of Landau's Fermi liquids. In spite of that, it has been recently shown[27] that in compressible normal metals a Luttinger surface as well as a Fermi surface give rise to similar Fermi liquid-like long-wavelength low-frequency linear response functions. In reality, a connection between Luttinger and Fermi surfaces was earlier noticed by Volovik[32,33], who showed that the Green's function close to both of them is characterised by a topological invariant that Heath and Bedell recently proved to be closely related to Luttinger's theorem[34].

Here we show that the relationship between Luttinger and Fermi surfaces is actually even much deeper, and entails the existence in both cases of 'quasiparticle' excitations whose lifetime grows to infinity approaching a Luttinger surface as well as a Fermi surface. Such ubiquitous 'quasiparticles' close to both Fermi and Luttinger surfaces, which yield further physical meaning to their common topological properties[32–34] and rationalise in simpler terms the results of ref. [27], may provide a new, broader paradigm for strongly correlated electron systems.

## Results

**Preliminary considerations.** We consider a generic non-insulating interacting electron system in three dimensions, whose single-particle Green's function (The Green's function is generically a matrix in the eigenbasis of the non-interacting Hamiltonian. Here we assume, for simplicity, that Green's function is diagonal on such a basis, although our final results do not depend on that choice. Moreover, we do not indicate explicitly the labelling of such eigenbasis, unless necessary.) of the complex frequency $\zeta$ can be written, through Dyson's equation, as

$$G(\zeta, \mathbf{k}) = \frac{1}{\zeta - \epsilon(\mathbf{k}) - \Sigma(\zeta, \mathbf{k})} = \int d\omega \, \frac{A(\omega, \mathbf{k})}{\zeta - \omega} \,, \qquad (1)$$

where $\epsilon(\mathbf{k})$ is the non-interacting dispersion measured with respect to the chemical potential, $\Sigma(\zeta, \mathbf{k})$ the self-energy, and $A(\omega, \mathbf{k}) > 0$ the single-particle DOS satisfying $\int d\omega \, A(\omega, \mathbf{k}) = 1$. We shall mostly work with the analytic continuation on the real axis from above, i.e. $G(\zeta = \epsilon + i\eta, \mathbf{k}) \equiv G_+(\epsilon, \mathbf{k})$ the retarded Green's function, and, similarly, $\Sigma(\zeta = \epsilon + i\eta, \mathbf{k}) \equiv \Sigma_+(\epsilon, \mathbf{k})$, with infinitesimal $\eta > 0$. For convenience, we absorb the Hartree–Fock contribution to the self-energy into a redefinition of $\epsilon(\mathbf{k})$ so that, by definition, $\Sigma_+(\epsilon \to \infty, \mathbf{k}) \to 0$.

One can readily show that order by order in perturbation theory the following result holds at $T = 0$:

$$\text{Im} \, \Sigma_+(\epsilon \to 0, \mathbf{k}) = -\gamma(\mathbf{k}) \, \epsilon^2 \,, \qquad (2)$$

with $\gamma(\mathbf{k}) > 0$, which is actually the starting point of the microscopic derivation of Landau–Fermi liquid theory[3,4].

However, the validity of Eq. (2) order by order in perturbation theory does not guarantee that the actual interaction strength, especially in strongly correlated electron systems, is within the convergence radius of the perturbation series, nor that the latter converges at all. Nonetheless, it has been recently shown[27] that Eq. (2) is not necessary but only sufficient to microscopically derive Landau-Fermi liquid theory.

**Key analytic assumptions.** Indeed, condition (2) refers just to the hypothetical decay rate of an electron undressed by self-energy corrections, which is not an observable quantity. On the contrary, for Landau's Fermi liquid theory to apply, we do have to impose a similar condition of vanishing decay rate, i.e. infinite lifetime, but for the actual quasiparticles, which reads, still at $T = 0$,

$$\lim_{\epsilon \to 0} \Gamma(\epsilon, \mathbf{k}) \equiv -\lim_{\epsilon \to 0} Z(\epsilon, \mathbf{k}) \, \text{Im} \, \Sigma_+(\epsilon, \mathbf{k}) = \gamma_*(\mathbf{k}) \, \epsilon^2 \,, \qquad (3)$$

where

$$Z(\epsilon, \mathbf{k})^{-1} \equiv 1 - \frac{\partial \, \text{Re} \, \Sigma_+(\epsilon, \mathbf{k})}{\partial \epsilon} \,, \qquad (4)$$

is the so-called quasiparticle residue, and $\gamma^*(\mathbf{k}) \geq 0$ provided $Z(\epsilon \to 0, \mathbf{k}) \geq 0$, which is generally true even though $Z(\epsilon, \mathbf{k})$ may well be negative at $\epsilon \neq 0$. Such physically sound replacement is actually the key of this work, and has huge implications, as we are going to show, despite it may look at first sight innocuous. For instance, Eq. (3) does include Eq. (2) when $Z$ is finite as a special case, but, e.g. remains valid even when $Z \sim \epsilon^2$ and $\text{Im} \, \Sigma_+$ is constant, the furthest possible case from a conventional Fermi liquid. We mention that in two dimensions the analytic $\epsilon^2$ dependence in Eq. (3) must be substituted by a non-analytic one[35], as non-analytic are the expected subleading

corrections to Eq. (3) also in three dimensions[36,37]. However, all the results we are going to derive are valid in three as well as in two dimensions. We further remark that while Eq. (2) is a perturbative result, Eq. (3) is a non-perturbative assumption based on the conjecture that 'quasiparticles' may exist also when perturbation theory breaks down. As such, Eq. (3) is not meant to describe generic non-Fermi liquids, as those classified in ref. [34], like marginal Fermi liquids[22] or Luttinger liquids[29]. For instance, in Luttinger liquids the right-hand side of Eq. (3) does vanish, but not continuously as $\epsilon \to 0$.

**Emergence of quasiparticles.** Let us now prove that Eq. (3) indeed admits the existence of 'quasiparticles'. We define

$$\Xi(\epsilon, \mathbf{k}) \equiv Z(\epsilon, \mathbf{k})\big(\epsilon(\mathbf{k}) + \mathrm{Re}\,\Sigma_+(\epsilon, \mathbf{k}) - \epsilon\big) = -\left(\frac{\partial \ln \mathrm{Re}\,G_+^{-1}(\epsilon, \mathbf{k})}{\partial \epsilon}\right)^{-1},$$

(5)

so that the single-particle DOS can be formally written as

$$A(\epsilon, \mathbf{k}) = \frac{1}{\pi} Z(\epsilon, \mathbf{k}) \frac{\Gamma(\epsilon, \mathbf{k})}{\Xi(\epsilon, \mathbf{k})^2 + \Gamma(\epsilon, \mathbf{k})^2},$$

(6)

and, for $\epsilon \simeq 0$, a 'quasiparticle' DOS

$$A_{\mathrm{qp}}(\epsilon, \mathbf{k}) \equiv Z(\epsilon, \mathbf{k})^{-1} A(\epsilon, \mathbf{k}) = \frac{1}{\pi} \frac{\Gamma(\epsilon, \mathbf{k})}{\Xi(\epsilon, \mathbf{k})^2 + \Gamma(\epsilon, \mathbf{k})^2}.$$

(7)

Since $\Gamma(\epsilon \to 0, \mathbf{k}) \to 0$, $A_{\mathrm{qp}}(\epsilon, \mathbf{k})$ generically vanishes for $\epsilon \to 0$ unless

$$E(\mathbf{k}) \equiv \lim_{\epsilon \to 0} \Xi(\epsilon, \mathbf{k}),$$

(8)

vanishes, too. That, because of the definition (5) of $\Xi(\epsilon, \mathbf{k})$, occurs only if $E(\mathbf{k})$ crosses zero. Therefore, in the present formalism there exists a unique surface in $\mathbf{k}$-space, which we dub as Fermi–Luttinger surface, where $\mathbf{k} = \mathbf{k}_{\mathrm{FL}}$ such that $E(\mathbf{k}_{\mathrm{FL}}) = 0$. Moreover, a reasonable assumption given Eq. (5) is that $\Xi(\epsilon, \mathbf{k}_{\mathrm{FL}})$ vanishes linearly in $\epsilon$. Indeed, we can envisage two different scenarios:

(F)  $\epsilon(\mathbf{k}_{\mathrm{FL}}) + \mathrm{Re}\,\Sigma_+(0, \mathbf{k}_{\mathrm{FL}}) = 0$, so that

$$\mathrm{Re}\,G_+^{-1}(\epsilon \to 0, \mathbf{k}_{\mathrm{FL}}) = Z(0, \mathbf{k}_{\mathrm{FL}})^{-1} \epsilon, \; \Xi(\epsilon \to 0, \mathbf{k}_{\mathrm{FL}}) = -\epsilon.$$

(9)

This is actually the case of conventional Fermi liquids, where $\mathbf{k}_{\mathrm{FL}}$ belongs to the Fermi surface and the Green's function has a simple pole at $\epsilon = 0$.

(L)  the self-energy has a pole at $\epsilon = 0$, hence

$$\mathrm{Re}\,G_+^{-1}(\epsilon \to 0, \mathbf{k}_{\mathrm{FL}}) \simeq -\frac{\Delta(\mathbf{k}_{\mathrm{FL}})^2}{\epsilon}, \; \Xi(\epsilon \to 0, \mathbf{k}_{\mathrm{FL}}) = \epsilon.$$

(10)

In this case, $\mathbf{k}_{\mathrm{FL}}$ lies on the Luttinger surface, where Green's function crosses zero at $\epsilon = 0$.

We may make a further step forward and assume that $\Xi(\epsilon, \mathbf{k})$ has a regular Taylor expansion for small $\epsilon$ and $\mathbf{k} \simeq \mathbf{k}_{\mathrm{FL}}$, at least to leading order. If so,

$$\Xi\big(\epsilon \to 0, \mathbf{k} \simeq \mathbf{k}_{\mathrm{FL}}\big) \simeq E(\mathbf{k}) \mp \epsilon,$$

(11)

where the minus and plus signs refer, respectively, to the cases (F) and (L) above. We remark that this assumption is not verified in generic non-Fermi liquids[34] and Luttinger liquids[29], where $\Xi(\epsilon, \mathbf{k})$ is non-analytic as $\epsilon \to 0$ and $\mathbf{k} \to \mathbf{k}_{\mathrm{FL}}$, and thus does not admit a regular Taylor expansion.

If we assume the analytic behaviour of Eqs. (3) and (11), we readily find that for small $\epsilon$ and close to the Fermi-Luttinger surface

$$A_{\mathrm{qp}}(\epsilon, \mathbf{k}) \simeq \frac{1}{\pi} \frac{\gamma_*(\mathbf{k})\epsilon^2}{(E(\mathbf{k}) \mp \epsilon)^2 + \gamma_*(\mathbf{k})^2 \epsilon^4} \simeq \delta(\epsilon \mp E(\mathbf{k})) \equiv \delta\big(\epsilon - \epsilon_{\mathrm{qp}}(\mathbf{k})\big).$$

(12)

That is exactly what we expect when approaching the Fermi surface in a conventional Fermi liquid, here valid also upon approaching the Luttinger surface. In that case the existence of a 'quasiparticle', with $\delta$-like DOS and dispersion $\epsilon_{\mathrm{qp}}(\mathbf{k}) = -E(\mathbf{k})$ in momentum space is in striking contrast with the behaviour of the experimentally accessible DOS of the physical electron, $A(\epsilon, \mathbf{k})$ in Eq. (6), which has instead a pseudogap since $Z(\epsilon \to 0, \mathbf{k}_{\mathrm{FL}}) \sim \epsilon^2$.

**Fermi liquid properties.** Since in both cases Eq. (3) holds, one can derive microscopically a Landau–Fermi liquid theory[27], and, correspondingly, a kinetic equation for the 'quasiparticle' distribution function, which looks exactly like the conventional one[1,2,4], apart from the fact that the 'quasiparticle' dispersion in momentum space is $\epsilon_{\mathrm{qp}}(\mathbf{k}) = E(\mathbf{k})$ and $\epsilon_{\mathrm{qp}}(\mathbf{k}) = -E(\mathbf{k})$ for cases (F) and (L), respectively. For instance, the specific heat can be calculated through the heat density–heat density response function, using the Ward–Takahashi identity[38]. We find, at leading order in $T$[39], that

$$c_V = -\frac{1}{V} \sum_{\mathbf{k}} \int \frac{d\epsilon}{T} \frac{\partial f(\epsilon)}{\partial \epsilon} (\Xi(\epsilon, \mathbf{k}) + \epsilon)^2 A_{\mathrm{qp}}(\epsilon, \mathbf{k})$$
$$\simeq \frac{\pi^2}{3} \frac{T}{V} \sum_{\mathbf{k}} \delta\big(\epsilon_{\mathrm{qp}}(\mathbf{k})\big) \equiv \frac{\pi^2}{3} T A_{\mathrm{qp}}(0),$$

(13)

hence the specific heat is still linear in $T$ even in Luttinger's case (L), despite the single-particle DOS pseudogap, a result which is also striking. Therefore, a pseudogap in the single-particle spectrum measured in ARPES and a finite specific heat coefficient $c_V/T$ are indirect evidences of those 'quasiparticles', although other explanations are well possible. Similarly, we expect that the coherent 'quasiparticles' will give rise to well-defined quantum oscillations in both Fermi and Luttinger surfaces, although the amplitudes might be different in the two cases. We remark that a finite $c_V/T$ in the case of quasiparticles close to a Luttinger surface entails singular vertex corrections that compensate the vanishing quasiparticle residue $Z$ to enforce the Ward–Takahashi identity.

Since Galilean invariance is lost on a lattice, one could in principle imagine a circumstance where the Drude peak vanishes despite a finite quasiparticle DOS at zero energy, thus a linear in temperature specific heat, and still compatible with Landau's Fermi liquid theory. Such anomalous situation, as many others one could think of inspired by the earlier mentioned phenomenology of some correlated materials, might occur more likely for quasiparticles at a Luttinger surface because of the singular vertex corrections. Moreover, an almost insulating behaviour in the charge transport as opposed to, e.g. a conventional Fermi liquid thermal one is hard to imagine when the single-particle DOS shows a quasiparticle peak but sounds more plausible with a pseudogap.

**Counting electron numbers through quasiparticle one.** In conventional Fermi liquids, Luttinger's theorem[31,40,41] states that the total number $N$ of particles at fixed chemical potential is simply the total number of quasiparticles, namely, considering, e.g. a single band of electrons and spin $SU(2)$ symmetry,

$$N = \sum_{\mathbf{k}} \sum_{\sigma=\uparrow,\downarrow} f\big(\epsilon_{\mathrm{qp}}(\mathbf{k})\big),$$

(14)

where $f(\epsilon) = \theta(-\epsilon)$ is the Fermi distribution function at $T = 0$. This result is just another manifestation of Landau's adiabatic hypothesis[1,2]. It is therefore worth showing how Eq. (14) changes when quasiparticles lie close to a Luttinger surface, namely, when

the interacting system is not adiabatically connected to the non-interacting one.

We recall that Luttinger's theorem requires just the existence of the Luttinger–Ward functional[42], and holds, in the more general sense outlined in the Supplementary Notes 1, provided at $T = 0$

$$\lim_{\epsilon \to 0} \text{Im}\, \Sigma_+(\epsilon, \mathbf{k})\, \text{Re}\, G_+(\epsilon, \mathbf{k}) = \lim_{\epsilon \to 0} \Xi(\epsilon, \mathbf{k})\, A_{\text{qp}}(\epsilon, \mathbf{k}) = 0\,, \tag{15}$$

which is verified for both (F) and (L) quasiparticles. Since the Luttinger–Ward functional can be constructed fully non-perturbatively[44], Luttinger's theorem remains valid beyond perturbation theory with the caveat highlighted in Supplementary Notes 1. Specifically, when perturbation theory breaks down leading to the appearance of a Luttinger surface, and thus of an additional zero of $\text{Re}\, G(i\epsilon, \mathbf{k})$ for $\epsilon \geq 0$, the expression of the total electron number may be different from that one deriving from conventional Luttinger's theorem $(14)$[43].

As discussed in Supplementary Notes 1, if we assume that perturbation theory may break down because of the proximity to a half-filled Mott insulator, Eq. (15) must be replaced by[43]

$$N = \frac{1}{2} \sum_{\mathbf{k}\sigma} \left( f(E(\mathbf{k})) + f(\epsilon(\mathbf{k})) \right). \tag{16}$$

Eq. (16) reduces to Eq. (14) when there is an adiabatic continuation, thus $E(\mathbf{k})$ and $\epsilon(\mathbf{k})$ have the same sign, which, not surprisingly, corresponds to case (F) with $\epsilon_{\text{qp}}(\mathbf{k}) = E(\mathbf{k})$, while it is different when perturbation theory breaks down at $\mathbf{k}$ and $E(\mathbf{k})\epsilon(\mathbf{k}) < 0$, case (L) with $\epsilon_{\text{qp}}(\mathbf{k}) = -E(\mathbf{k})$. Let us assume that the system is not far from the point in the Hamiltonian parameter space where perturbation theory breaks down and that $N$ at fixed chemical potential is smooth crossing that point. Under that assumption, since $N = \sum_{\mathbf{k}\sigma} f(\epsilon(\mathbf{k}))$ on the side where adiabatic continuation holds true, then Eq. (16) implies that on either side

$$N = \sum_{\mathbf{k}\sigma} f(\epsilon(\mathbf{k})) = \sum_{\mathbf{k}\sigma} f(E(\mathbf{k})) = \sum_{\mathbf{k}\sigma} \begin{cases} f\left(\epsilon_{\text{qp}}(\mathbf{k})\right) & \text{cases (F)}, \\ 1 - f\left(\epsilon_{\text{qp}}(\mathbf{k})\right) & \text{cases (L)}. \end{cases} \tag{17}$$

In other words, the quasiparticle 'Fermi' surface encloses a volume fraction of the whole Brillouin zone, $\mathbf{k}: \epsilon_{\text{qp}}(\mathbf{k}) < 0$, equal to the electron filling fraction $\nu$ in case (F), and the complement hole fraction $1 - \nu$ in case (L).

**Physics of a toy self-energy.** To make our point clearer, we present an explicit example based on a toy self-energy vaguely inspired by the phenomenology and by model calculations for the pseudogap phase of underdoped cuprates [45–52].

Assume a two-dimensional (2D) square lattice, a less than half-filled band with non-interacting dispersion $\epsilon(\mathbf{k})$ that gives rise to a closed Fermi surface, see Fig. 1, and a model self-energy[45] at very small $\epsilon$

$$\Sigma_+(\epsilon, \mathbf{k}) = \frac{\Delta(\mathbf{k})^2}{\epsilon + \epsilon_*(\mathbf{k}) + i\, \gamma(\mathbf{k})\, \epsilon^2}\,, \tag{18}$$

which, because of the imaginary term in the denominator, does satisfy Eq. (3) for any $\mathbf{k}$ (Strictly speaking, in two dimensions the $\epsilon^2$ dependence in Eqs. (2) and (3) should be replaced by $-\epsilon^2 \ln \epsilon$. Here, we still assume $\epsilon^2$, as in a quasi 2D model while neglecting a weak dispersion along the third dimension.)

$$\Sigma_+(\epsilon, \mathbf{k}) = \frac{\Delta(\mathbf{k})^2}{\epsilon + i\, \gamma(\mathbf{k})\, \epsilon^2} \underset{\epsilon \to 0}{\simeq} \frac{\Delta(\mathbf{k})^2}{\epsilon} - i\, \Delta(\mathbf{k})^2\, \gamma(\mathbf{k})\,,$$

is highly singular. We also assume, again unlike real systems, $\Delta(\mathbf{k}) = \Delta$ and $\gamma(\mathbf{k}) = \gamma$ to be independent of $\mathbf{k}$. We thus readily find

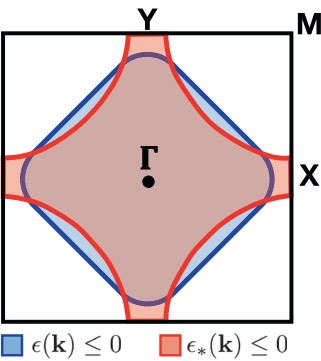

$\square\ \epsilon(\mathbf{k}) \leq 0 \qquad \square\ \epsilon_*(\mathbf{k}) \leq 0$

**Fig. 1 Luttinger surface of the model self-energy.** In black, the Brillouin zone with the high symmetry points. The non-interacting Fermi area, $\epsilon(\mathbf{k}) \leq 0$, is drawn in blue, while the interacting Luttinger one, $\epsilon^*(\mathbf{k}) \leq 0$, is in red. They are obtained, respectively, assuming $\epsilon(\mathbf{k}) = -2\cos k_x - 2\cos k_y - \mu$, with $\mu = -0.2$, and $\epsilon^*(\mathbf{k}) = -2\cos k_x - 2\cos k_y + 4\, t'\, \cos k_x \cos k_y - \mu^*$, with $t' = 0.3$ and $\mu^*$ such that both areas are equal, thus forcing by hand the same number of electrons in the non-interacting and interacting cases.

that

$$E(\mathbf{k}) = \epsilon_*(\mathbf{k})\, \frac{\epsilon_*(\mathbf{k})\, \epsilon(\mathbf{k}) + \Delta^2}{\Delta^2 + \epsilon_*(\mathbf{k})^2}\,, \tag{19}$$

vanishes at $\epsilon^*(\mathbf{k}) = 0$, which defines the Luttinger surface, case (L) above, and at $\epsilon^*(\mathbf{k})\, \epsilon(\mathbf{k}) = -\Delta^2$, provided the latter equation admits real roots, which would then belong to case (F). Let us first assume $\epsilon^*(\mathbf{k})\, \epsilon(\mathbf{k}) + \Delta^2 > 0$ throughout the Brillouin zone, so that $E(\mathbf{k})$ just vanishes on the Luttinger surface $\mathbf{k} = \mathbf{k}_L$ with $\epsilon^*(\mathbf{k}_L) = 0$, close to which $E(\mathbf{k}) \simeq \epsilon^*(\mathbf{k})$. Making the same assumption that leads to Eq. (17), the Luttinger volume in this case comprises all $\mathbf{k}$ such that $\epsilon^*(\mathbf{k}) \leq 0$, which we suppose give rise to an open Luttinger surface, contrary to the closed non-interacting Fermi surface, both shown in Fig. 1.

The equation $\epsilon = \epsilon(\mathbf{k}) + \text{Re}\Sigma_+(\epsilon, \mathbf{k})$ has two roots, $\epsilon = \epsilon_-(\mathbf{k}) < 0$ and $\epsilon = \epsilon_+(\mathbf{k}) > 0$. Since

$$-\text{Im}\, \Sigma_+(\epsilon, \mathbf{k}) = \frac{\Gamma\, \Delta^2\, \epsilon^2}{\left( \epsilon + \epsilon_*(\mathbf{k}) \right)^2 + \gamma^2\, \epsilon^4}\,, \tag{20}$$

is peaked at $\epsilon = -\epsilon^*(\mathbf{k})$, the physical particle DOS, $A(\epsilon, \mathbf{k})$, displays two asymmetric peaks at $\epsilon = \epsilon_\pm(\mathbf{k})$, blue in Fig. 2. The negative energy peak is higher than the positive energy one when $\mathbf{k}$ is inside the Luttinger surface, i.e. $\epsilon^*(\mathbf{k}) < 0$, and the opposite when $\mathbf{k}$ is outside, much the same as for a conventional Fermi surface, despite here $A(\epsilon, \mathbf{k}) \sim \epsilon^2$ vanishes quadratically at $\epsilon = 0$. The low-energy quasiparticle DOS, $A_{\text{qp}}(\epsilon, \mathbf{k})$, red in Fig. 2, shows a peak that sharpens approaching the Luttinger surface, and moves oppositely from a conventional Fermi liquid: inside the Luttinger surface, the peak is at positive energy, while at negative energy outside.

We note that, according to Eq. (16), the presence of only a pseudo-gapped Luttinger surface without Fermi pockets implies an electron density stuck to half-filling, thus an incompressible state, and, yet, supporting 'quasiparticles', i.e. gapless fermionic excitations; a remarkable physical situation if it were ever realised.

Assume now that the equation $\epsilon(\mathbf{k})\, \epsilon^*(\mathbf{k}) = -\Delta^2$ admits two real roots, which create small Fermi pockets in the regions where the Luttinger and the non-interacting Fermi surfaces do not overlap, shown in green in Fig. 3. Since these roots belong to case (F), the physical particle DOS should develop peaks at $\epsilon = 0$ along the borders of such pockets. However, the pseudogap on the Luttinger surface implies that the peaks along the arcs closer to the non-interacting Fermi surface, bold green lines in Fig. 3, are

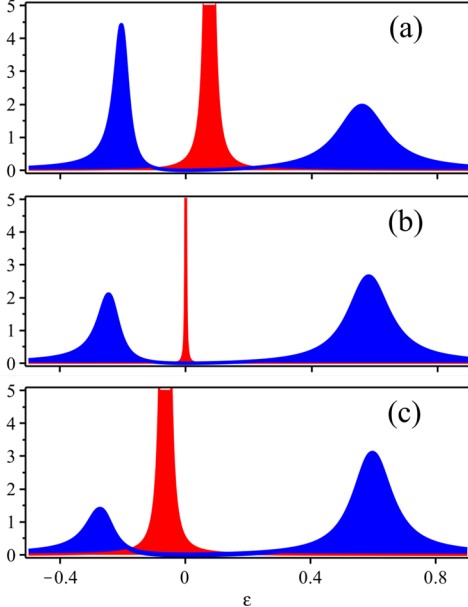

**Fig. 2 Spectral functions crossing the Luttinger surface.** Physical particle, $A(\epsilon, \mathbf{k})$ in blue, and 'quasiparticle', $A_{qp}(\epsilon, \mathbf{k})$ in red, densities of states for $\mathbf{k}$ along the path $\mathbf{Y} \to \mathbf{M}$. **a**–**c** refer, respectively, to $\mathbf{k} = (0.8\, k_L, \pi)$, $\mathbf{k} = (k_L, \pi)$ on the Luttiger surface, and $\mathbf{k} = (1.1\, k_L, \pi)$. We use $\epsilon(\mathbf{k})$ and $\epsilon_*(\mathbf{k})$ as in Fig. 1 and take $\Delta = 0.4$, so that $\epsilon(\mathbf{k})\, \epsilon_*(\mathbf{k}) + \Delta^2 > 0$ throughout the Brillouin zone, and $\gamma = 1$. We also added a finite broadening to make the quasiparticle visible on the Luttinger surface.

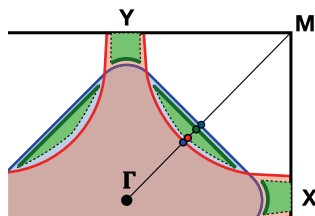

**Fig. 3 Coexisting Luttinger surface and Fermi pockets.** Fermi pockets, in green, which arise when the equation $\epsilon(\mathbf{k})\, \epsilon_*(\mathbf{k}) = -\Delta^2$ admits real roots. The dots along the $\mathbf{\Gamma} \to \mathbf{M}$ direction are the points at which we calculate the single-particle DOS in Fig. 4. The bold green arcs along the Fermi pockets boundaries are the positions in $\mathbf{k}$-space of the highest zero-energy peaks in the physical electron DOS.

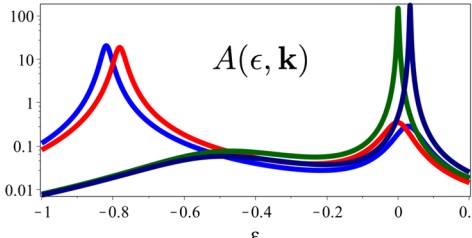

**Fig. 4 Spectral function crossing Luttinger and Fermi surfaces.** Physical electron DOS with the same parameters as in Fig. 2 apart from $\Delta = 0.1$, a value at which $\epsilon(\mathbf{k})\, \epsilon_*(\mathbf{k}) = -\Delta^2$ has real solutions. The different curves refer to the $\mathbf{k}$-points along $\mathbf{\Gamma} \to \mathbf{M}$ shown in Fig. 3. Specifically, the blue curve is on the Luttinger surface, the red at the inner border of the Fermi pocket, the dark green at the outer border, and, finally, the dark blue on the non-interacting Fermi surface. Note the much greater height of the zero-energy peak for $\mathbf{k}$ close to the non-interacting Fermi surface than to the Luttinger one.

much more pronounced. This is explicitly shown in Fig. 4, where we draw the physical particle DOS on the points along $\mathbf{\Gamma} \to \mathbf{M}$ of Fig. 3.

In other words, moving from the case in which $\epsilon(\mathbf{k})\, \epsilon^*(\mathbf{k}) = -\Delta^2$ has no solution to that in which the solution exists, our toy self-energy describes a kind of Lifshitz's transition resembling that observed by cluster extensions of dynamical mean field theory in the 2D Hubbard model upon increasing hole doping away from half-filling[46,51,52], which in turn has been associated with the Fermi surface evolution across the critical hole-doping level at which the pseudogap vanishes.

We mention that $E(\mathbf{k})$ of Eq. (8) along, e.g. the path $\mathbf{\Gamma} \to \mathbf{M}$ in Fig. 3 crosses three zeros, which may evoke the so-called 'fermion condensation'[53,54]. However, in the present case, two of the three zeros refer to divergences of $G_+(\epsilon, \mathbf{k})$, the boundaries of the Fermi pocket in Fig. 3, while the third to a zero of $G_+(\epsilon, \mathbf{k})$, the Luttinger surface, which avoids the band flattening mechanism at the fermion condensation[55].

**Discussion.** We have shown that coherent 'quasiparticles' emerge both approaching Fermi *and* Luttinger surfaces. This result expands the class of interacting electron systems, like, e.g. Luttinger liquids[29], which are predicted to display conventional Landau–Fermi liquid behaviour, despite very different and anomalous single-particle properties. Moreover, it implies that also close to a Luttinger surface the low-energy physics may possess the huge emergent symmetry[56] recently discussed in great detail by ref. [28], which is remarkable given the single-particle density pseudogap at the Luttinger surface.

## Data availability

The author declares that all data supporting the findings of this study are available within the paper and its Supplementary Information files.

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

## Acknowledgements

I am grateful to Erio Tosatti, Antoine Georges, Marco Schirò, Grigory Volovik, and Jan Skolimowski for helpful discussions and comments. This work has received funding from the European Research Council (ERC) under the European Union's Horizon 2020 research and innovation programme, Grant agreement No. 692670 "FIRSTORM".

## Competing interests

The author declares no competing interests.
