## [Peer Review File · Nature Communications]

Reviewers' Comments:

Reviewer #1:

Remarks to the Author:
please find the attached file

Referee Report for Nature Communications: Landau Fermi Liquids in Disguise Michele Fabrizio

This paper presents an interesting perspective on how Fermi Liquid (FL) like properties can be realized by Green's function with different analytic properties connected to interacting fermions with a Fermi Surface or a Luttinger Surface. One of the major properties of a Fermi liquid is Luttinger's Theorem (L'sT) and for a long time there was a belief that in 1D L'sT did not work for the Luttinger Liquid (LL). The first papers to show that L'sT does work for the 1D LL were two PRL's, 1) and 2), below. In the Sólyom reference, 3) below, is a comprehensive review of 1D systems. In Sólyom's review (one of the best of many reviews) he shows some of the thermodynamic properties, in particular the specific heat of a 1D LL that looks like a 1D FL. I understand that the calculations in Fabrizio's paper are in 2D and 3D and I know that it is hard to find exact solutions as in the 1D examples but it might be useful to see how this approach works in the 1D world.

I have two concerns about the paper. The first one is that the mathematical derivation from page 2 – page 4 should be in the Supplementary section and only the basic results should be in the main paper. From page 5 – page 10 with a simpler derivation and with the few figures the explanation was clearer. I say this because I have been doing calculations of this sort for decades and I still had trouble following the details of the calculation. I would have preferred to see the results along with some more discussion about how these results might add something new or provide new insights to some of the work in Ref. 1-7, in Fabrizio's paper. The second one is that I am not sure if Fabrizio has read the derivation of L'sT in Ref. 4 below. The derivation in 4) is not perturbative, it only depends on the ratio of the (Real part of the Green's function/Imaginary part of the Green's Function) $\rightarrow 0$. The Green's function approximations Fabrizio used would not violate L'sT and they would be non-perturbative. With his assumption for the Green's functions his calculations would be valid beyond perturbation theory as seen in Ref. 4 below.

- 1) Blagoev and Bedell, Phys. Rev. Lett. **79**, 1106 (1997).
- 2) Yamanaka, Oshikawa, and Affleck, Phys. Rev. Lett. **79**, 1110 (1997).
- 3) J. Sólyom, Adv. Phys. **28**, 201 (1979).
- 4) Abrikosov, Gorkov, & Dzyaloshinski, Sec. 19.4 (Dover Paper Back, 1975).

Reviewer #2:

Remarks to the Author:

It is conventional accepted that the well defined quasi-particle excitation does not exist when approaching the Luttinger surface in the correlating electron system. However, by imposing a condition of vanishing decay rate, the author show that finite coherent quasi-particle exists at both the Fermi and Luttinger surface, quantum oscillation of the well defined quasi-particle is yielded at the Luttinger surface, in addition with linear specific heat. The theoretic predication might extends the subclass of correlated electron system. The work is novel and creative, and the theoretical deriv is rigorous, I have several suggestions and comments, for the manuscript is written highly specialized in theoretic condensed matter physics, improvement should be made for appealing a wider readers in Nat. Comm.

1. A more comprehensive introduction is needed, including more details about the progress in theoretic and experimental studies of Landau-Fermi liquid related to this work.

2. Would the well defined quasi-particle at the Luttinger surface be possibly observed experimentally, such as by ARPES (Angular resolved photoelectron spectroscopy), please comment this.

3. Please comment about whether the quasi-particle at the Luttinger surface have Anomalous effect in the conductivity, charge or heat transport in the predicted correlated electron liquid.

Reviewer #3:

Remarks to the Author:

Referee report on "Landau-Fermi liquids in disguise" by Michele Fabrizio

In this work, the author presents a phenomenological approach towards generalising the well known Landau Fermi liquid paradigm. This is done by relaxing the stringent condition of the existence of a single particle spectral peak at the Fermi energy, and proposes to focus instead on the weaker condition of the existence of long-lived excitations proximate to (i) the Fermi surface of solutions for the zeros of the inverse Greens function, or (ii) the Luttinger surface of solutions of the zeros of the Greens functions.

Further, the author assumes in equation (6) of the work that the decay rate of the quasiparticle excitations (which are not necessarily the physical electrons or even the Landau quasiparticle excitations of the Fermi liquid) possesses the same phenomenological form of that known from Landau Fermi liquid theory, i.e., the decay rate $\Gamma \sim \epsilon^2$, where ϵ is the energy scale of the quasiparticle excitations. This assumption then forms the cornerstone for the rest of the work.

The author then presents general arguments for the behaviour of the quasiparticle spectral function at Fermi as well as Luttinger surfaces. While the former is well known from the Fermi liquid phenomenology, the author studies the case of Luttinger surfaces that arise from a pole of the single-particle self-energy at zero frequency. The author then shows that while the residue of the physical electrons have a pseudogap (i.e., vanishes smoothly and continuously at vanishing frequency), a suitably defined quasiparticle spectral function has a delta-function behaviour at the Luttinger surface.

This suggests the existence of gapless excitations proximate to the Luttinger surface that satisfy the requirements of the Landau Fermi liquid phenomenology. Relying on an earlier work of the author (Ref. 7 of the manuscript), the author argues that such coherent quasiparticles give rise to various experimentally measurable quantities such as the "electronic" specific heat and quantum oscillations that have the precise forms known from a quantum Boltzmann transport equation formalism applied to Fermi liquid theory.

Finally, the author presents the case of a toy model single-particle self-energy inspired by DMFT calculations for the pseudo-gap phase of the underdoped cuprate materials (as well as a certain

quantum impurity model studied by the author and his collaborators, whose works are referenced in Ref.7 of this work). In the present work, the author shows that the assumption of equation (6) is satisfied by the model self-energy, and that the quasiparticle spectral function does show a delta function near the Luttinger surface (as well as possible poles of the Green function for the physical electrons near a Fermi surface).

Plots of these spectral functions for the quasiparticle and the physical electrons are presented in Figs.3 and 5, displaying the different kinds of low energy excitations lying near the Fermi and Luttinger surfaces respectively along the nodal direction of the underlying 2D tight-binding Brillouin zone (and passing through a Fermi pocket). The phenomena seen in these plots are reminiscent of the Lifshitz transition observed in dynamical cluster approximation studies of the hole-doped 2D Hubbard model. While the rest of the present work (formalism, toy model under discussion) borrows heavily from the author's earlier work (Ref. 7), plots 3 and 5 are the truly new results offered here.

While the formalism and discussions related to the toy model are a bold attempt at understanding whether the phenomena often ascribed to non-Landau Fermi liquid behaviour (observed in pseudogapped phases in models of strongly correlated electronic systems as well as materials) can instead be considered as falling within the Landau Fermi liquid paradigm, I have a couple of questions that must be addressed in order to ascertain the correctness, as well as a deeper understanding, of the work.

Comment 1. There appears to be a deep contradiction between the conclusions that arise from the present work and those obtained from the recent work by Heath and Bedell (Ref. 3 of the manuscript). Heath et al. offer a mathematically rigorous and in-depth analysis of the necessary and sufficient conditions that govern the validity of Luttinger's theorem, as well as the existence of gapless excitations near Fermi and Luttinger surfaces. Namely, they show that as long as the single-particle self-energy is analytic everywhere in the complex frequency plane (such that the Fermi and Luttinger surfaces are neither gapped nor pseudogapped), both types of surfaces (i.e., Fermi and Luttinger) are characterised by a non-zero topological index. This index corresponds to a winding number obtained by encircling a phase singularity of the Baym-Kadanoff generating functional that exists around these surfaces.

This topological index is shown to be equivalent to that proposed by Volovik (Ref. 1 of the manuscript). Further, the authors also establish by using rigorous arguments from algebraic topology (e.g., the Atiyah-Singer index theorem) that the existence of such a non-zero topological index guarantees the existence of gapless chiral excitations (proximate to the Fermi/Luttinger surface) whose dynamics can be studied in terms of an independent particle approximation.

This reveals an important contradiction with the assumptions and findings of the present work. First, in a significant departure from the assumptions of Heath et al., the author of the present work considers the existence of a pole in the single particle self-energy at zero frequency (i.e., a non-analyticity of the self-energy at the origin of the complex frequency plane) while studying the case of a Luttinger surface with pseudogapped physical electrons. Thus, the finding of gapless quasiparticle excitations proximate to such a Luttinger surface is in direct conflict with the outcome obtained from the analysis of Heath et al. mentioned above. Indeed, Heath et al. appear to rule out such a possibility for a pseudogapped Luttinger surface.

Further, the results of Heath et al. show that such a pseudogapped Luttinger surface will clearly violate Luttinger's theorem. Yet, the author of the present work appears to be assuming by hand that Luttinger's theorem continues to be valid in the presence of such a pseudogapped Luttinger surface (see, e.g., the end of the caption of Figure 2). Again, this is in clear contradiction with the rigorous results of Heath et al. These contradictions cast considerable doubt on the correctness of the approach taken in this work, and need to be resolved urgently.

Comment 2. Further, the work by Heath et al. lays the foundations by which to show the conserved Luttinger volume (i.e., the validity of Luttinger's theorem) for all non-Landau Fermi liquids whose gapless excitations lie proximate to a Luttinger surface. The Tomonaga-Luttinger liquid in 1D and the Marginal Fermi liquid (MFL) phenomenology in 2D are studied as particular examples, but very general criteria are also laid out for non-Landau Fermi liquids beyond these two well known cases. A similar result has also been obtained for the MFL in A. Mukherjee et al., Nucl. Phys. B 960, 115170 (2020).

While the present work appears to make a case for the existence of a generalised Landau Fermi liquid phenomenology, this is clearly not true for the case of the MFL. From the well known phenomenological self-energy of the MFL, it can be easily seen that the decay rate of the single-particle excitations do not satisfy the assumption made in equation (6) of the present work: $\Gamma_{\text{MFL}} \sim (\epsilon / |\ln \epsilon|)$ instead of the $\Gamma \sim \epsilon^2$ assumed in equation (6). Thus, the MFL clearly does not belong to the generalised Landau Fermi liquid phenomenology that the author is attempting to formulate here. It appears very likely that many of the other non-Fermi liquids (lying proximate to Luttinger surfaces) classified by Heath et al. using topological arguments will also not fit within the formulation presented in this work.

This points towards a clear limitation of the main assumption, i.e., the form of the decay rate given in equation (6). It is, therefore, important for the author to carefully consider the qualifications that must be placed on equation (6). The fact that the toy model of a pseudogapped Luttinger surface agrees with equation (6) as presently formulated may well point to a special case of a self-energy (given above equation (23)) that fits within the requirements of the author's formulation of a generalised Landau Fermi liquid phenomenology. Given the evidence stated above that points to various non-Fermi liquids not falling within this phenomenology, I cannot at this point agree with the author's statement towards the end of the manuscript which states that "This result expands substantially the class of interacting electron systems ". This will need further in-depth study of various self-energies for gapless excitations lying near the Luttinger surface beyond the toy model considered here.

Summary: To conclude, while the author has made an interesting attempt at formulating a generalised Landau Fermi liquid phenomenology, I have raised some concerns above that question the correctness of the main assumption made in this work, as well as the results obtained therefrom. I believe that it is important for the author to address these concerns appropriately prior to my accepting the work as ready for publication.

Reply to Reviewer #1

I thank the Referee for her/his positive assessment of the work and useful comments. In what follows, I will reply point by points to those comments.

1. The Referee is perfectly right pointing out that in 1D Luttinger liquids Luttinger's theorem holds and, in addition, the low-energy long-wavelength behaviour of the response functions, thus of the uniform thermodynamic susceptibilities, are those of a conventional Landau's Fermi liquid, despite the absence of coherent quasiparticles. It is therefore perfectly legitimate to ask how the physics of Luttinger liquids compares with that I have discussed in the manuscript. The key point where my derivation deviates from what is known to occur in 1D is the assumption that $\Gamma(\epsilon, \mathbf{k})$ and $\Xi(\epsilon, \mathbf{k})$ defined, respectively, in Eqs. (6) and (9) of the original version, (3) and (5) of the revised one, have a regular Taylor expansion, at least to leading order, close to $\epsilon = 0$ and what I dub as Luttinger-Fermi momentum \mathbf{k}_{FL} . Such expansion fails in 1D since the self-energy is not analytic in a whole region, and not in a single point as in the case I discuss. I apologise for not having mentioned that important fact in the manuscript, which I remedy in the revised version. The consequence of that non-analytic behaviour is finally the absence of quasiparticles in 1D, i.e., of coherent single-particle excitations.

In spite of that, the dynamical properties of particle-hole excitations at low energy and momentum transferred are similar in both Fermi and Luttinger liquids, as exhaustively reviewed by Sólyom, Ref. [27] in the revised version, and explained by Dzyaloshinskii and Larkin back in 1974.

2. Just like the Referee, I also believe that the standard proof of Luttinger's theorem, as that outlined in the book by Abrikosov, Gorkov, and Dzyaloshinskii (AG&D), now Ref. [39], is essentially non perturbative. However, many authors do not agree with such statement because the Luttinger-Ward functional, X in AG&D book and Φ in the Supplementary section of the manuscript, is defined through a series expansion, though recasted in terms of skeleton diagrams. In fact, many examples have been discussed over the years where Luttinger's theorem seems violated, see, e.g., Refs. [57] to [60] of the revised version. Indeed, I have been for long strongly puzzled by those works. Finally, in the last few months my collaborator, Jan Skolimowski, and I have been able to prove that those claimed violations are untrue and derive from an improper use of Luttinger's theorem. In short, Luttinger's theorem predicts that the total number of electrons can be simply calculated through the total phase accumulated by $\prod_{\mathbf{k}\sigma} G_{\sigma}(i\epsilon, \mathbf{k})$ from $\epsilon = 0$ to $\epsilon = \infty$ (We work for convenience in Matsubara's formalism, since on the positive imaginary axis the Green's function has a regular analytic behaviour, see Dzyaloshinskii, Physical Review B **68**, 085113 (2003)). However, that phase is generally not simply the sum of the phases of each $G_{\sigma}(i\epsilon, \mathbf{k})$, since the single particles states (σ, \mathbf{k}) may be strongly entangled together

by interaction. Therefore, one needs to properly define that phase via an appropriate modulo operation. We have found that the proper choice is the one that makes the so-called Luttinger integral vanish, i.e.,

$$I_L = \int_0^\infty \frac{d\epsilon}{\pi} \sum_{\mathbf{k}\sigma} \text{Im} \left(G_\sigma(i\epsilon, \mathbf{k}) \frac{\partial \Sigma_\sigma(i\epsilon, \mathbf{k})}{\partial \epsilon} \right) = 0. \quad (1)$$

Indeed, any improper choice, as, e.g., the naïve one of choosing the sum of the phases, may lead to a finite I_L integer multiple of $1/2$, which is just the missing quantised term found in some of the aforementioned works. Remarkably, the simple sum of the phases does lead to $I_L = 0$ only when perturbation theory does not break down, i.e., Landau's adiabatic hypothesis holds true.

Therefore, according to us, the correct statement of Luttinger's theorem is that there always exists a proper modulo operator on the phase of $\prod_{\mathbf{k}\sigma} G_\sigma(i\epsilon, \mathbf{k})$ that makes $I_L = 0$, and thus allows calculating the total electron number via the accumulated phase from 0 to ∞ .

We are now completing a work that will include those results. However, in light of the role played by Luttinger's theorem in the manuscript, I have modified the text making reference to that unpublished work, Ref. [42] in the revised version, and just quoting the main outcome. As a result, a substantial part of the previous text has become outdated and unnecessary, so that, also following Referee's comment, it does not appear in the revised version. Instead, following other Referees' comments, I have added a more general and broader introduction, and exchanged the order of the derivation. Specifically, I first show that my assumptions about $\Gamma(\epsilon, \mathbf{k})$ and $\Xi(\epsilon, \mathbf{k})$ lead to the existence of quasiparticles at a Luttinger surface as well as at a Fermi surface (since the latter is just a special case, I have finally decided not to move what was earlier written in pages 2-4 in the Supplementary Notes), and next I show how Luttinger's theorem looks in that case. The reason of that exchange is that, reading one of the other Referee reports, I realised that somebody could erroneously conclude that my results rely on the validity of Luttinger's theorem, while I just wanted to show that they are compatible with that theorem.

I hope that the revised version and the discussion above satisfactory answer Referee's comments, whom I thanks again for her/his invaluable help in improving the manuscript.

Reply to Reviewer #2

I thank the Referee for her/his positive assessment of the work and useful comments. In what follows, I reply point by points to those comments.

1. I agree that the introduction was too technical in the original version, as the Referee points out. Therefore, following her/his suggestion, I added a broader and more comprehensive introduction where I discuss the physical motivations of the work, namely the evidence that in several correlated materials a well defined quasiparticle peak in the photoemission spectrum is absent and replaced by a pseudogap, which is conventionally interpreted as a manifestation of non-Fermi liquid behaviour, yet other physical properties are perfectly Fermi liquid-like.
2. The 'quasiparticles' emerging out of a Luttinger surface are characterised by a pseudogap in the single particle density of states. Therefore, they cannot be revealed in photoemission. The coexistence of a pseudo-gap in the ARPES spectrum with a linear specific heat and/or well defined quantum oscillations, could represent an indirect evidence of those 'quasiparticles'. I mentioned this aspect in the original version, but maybe not stressing it enough, which I do in revised one.
3. This is a very important point. One can derive from the 'quasiparticles' close to the Luttinger surface a kinetic equation exactly like one does for conventional quasiparticles. However, I believe the existence of such 'quasiparticles' crucially relies on an underlying lattice breaking Galilean invariance. Therefore, for instance, the Drude peak would involve a Landau's f -parameter which one cannot fix by Ward identities. It could be well possible that the net result is a vanishingly small Drude peak despite a sizeable specific heat coefficient, thus something that looks almost like an insulator from the point of view of charge transport, but with metallic specific heat, and, eventually, finite paramagnetic susceptibility and/or well defined quantum oscillations. Such anomalous behaviour is actually not incompatible with Landau's kinetic equation once Galilean invariance is lost even for conventional quasiparticles, but becomes all the more plausible when even the single-particle spectrum looks almost that of an insulator.

This is just a speculative scenario, which I cannot justify by any means, though it is rather suggestive, having in mind striking evidences as the observation of quantum oscillations characteristic of a large Fermi surface in the insulator SmB_6 , see Suchitra Sebastian *et al.* in *Science* **349**, 287 (2015).

I was originally a bit reluctant to put forward such speculative scenarios, which I could not support by a model calculation. However, in light of Referee's comment, I have decided to mention it in the revised version.

In addition, meeting other Referees' requests, I have modified the order of presentation of the results. Specifically, I first show that my assumptions about $\Gamma(\epsilon, \mathbf{k})$ and $\Xi(\epsilon, \mathbf{k})$ lead to

the existence of quasiparticles at a Luttinger surface, and next I show how Luttinger's theorem looks in that case. The reason of that exchange is that, reading one of the other Referee reports, I realised that somebody could erroneously conclude that my results rely on the validity of Luttinger's theorem, while I just wanted to show that they are compatible with that theorem. Moreover, in the meanwhile we have found some new results about Luttinger's theorem showing, e.g., that it holds even when was claimed not to. Therefore, in the revised version I just mention the outcome of that work, Ref. [42], which we will soon post on the web, and use it, though the final result that the number of quasiparticles at a Luttinger surface counts the number of physical holes, while that at the Fermi surface the number of physical particles remains unchanged. In this way, I could erase all technical details about Luttinger's theorem, which are now outdated and unnecessary.

I hope that the revised version and the discussion above satisfactory answer Referee's comments, whom I thanks again for her/his invaluable help in improving the manuscript.

Reply to Reviewer #3

I thank the Referee for carefully reading the manuscript and for rising pertinent and useful comments that helped me improving the presentation in the revised version. In what follows, I will reply point by points to those comments.

1. The Referee is correct in pointing out similarities and discrepancies between the results in the manuscript and those in the nice work by Heath and Bedell. Let me first point out the common results, and next the discrepancies.
 - Heath and Bedell states in Corollary 1.1 that *The topological index of the D-dimensional generating functional for all two-point Green's functions cannot distinguish between the presence of a (D-1)-dimensional Fermi surface and a (D-1) dimensional Luttinger surface*, and, after Eq. (16), that "the presence of Fermi/Luttinger surfaces in fermionic matter is equivalent to the appearance of an anomaly in the quantized many-body field theory", which, according to them, is associated with the existence of gapless excitations, as they discuss after Corollary 1.2. Specifically, they state that such anomaly leads to "a non-zero condensate of particle-hole pairs with a linearized dispersion", which, maybe erroneously, I interpret as the conventional Fermi liquid-like particle-hole density of states that vanishes linearly at zero energy. Finally, in Theorem 2. they state that *A non-zero value for the topological index of a D-dimensional Kadanoff-Baym functional is the sole necessary and sufficient condition for the validity of Luttinger's theorem.*
All together the above statements seem to agree with the results I obtain. Indeed, I explicitly show that even when a Luttinger surface exists, on which the single-particle density-of-states vanishes quadratically at zero energy, still the specific heat is linear in temperature, signal of particle-hole excitations with linearly vanishing density of states. I emphasise that the proof of the latter result, Eq. (21) of the original version, does not require Luttinger's theorem but just the Ward-Takahashi identity.
 - Despite those close similarities, Heath and Bedell conclude in Corollary 2.1 that *The only possible scenario where Luttinger's theorem fails is in the presence of a gap or pseudogap*, in contrast to what I claim.

Honestly, I could not understand how Corollary 2.1 arises and is compatible with what stated in Theorem 2. Luttinger's theorem relies on the existence of a Luttinger-Ward functional, which, as Pothoff has shown in *Condens. Mat. Phys.* **9**, 557 (2006), can be derived fully non-perturbatively. Therefore, I cannot think of any reason why that theorem should not work in presence of a gap or a pseudogap, apart from a zero temperature instability as argued by Dzyaloshinskii, *Physical Review B* **68**, 085113 (2003), and by Pothoff himself. Therefore, although I do agree with most of the results obtained by Heath and Bedell, I

disagree with Corollary 1.2. In fact, we are now completing a work where we show that all examples present in the literature that have been invoked as clear violations of Luttinger's theorem due to singularities in the self-energy leading to a true or a pseudo gap, as, for instance, Refs. [40] and [115] in the work by Heath and Bedell, or Refs. [57] to [60] in the revised manuscript, do not actually violate that theorem. Since the demonstration works perfectly well also in those examples that refer to a Mott insulator with a hard gap, like in Ref. [58] of the revised version, there is evidently something missing in Corollary 2.1 that I cannot identify. Since I think it is unfruitful to open a controversy, given that my results agree with most of Heath&Bedell's ones, I prefer not to highlight in the revised version the contradictions pointed out by the Referee, as I intentionally did not in the original version. However, in the revised version I refer explicitly to that unpublished work, Ref. [42], and briefly mention the outcome, which allows me to substantially shorten the initial part, unnecessary in light of that result, and thus to extend the introductory physical discussion.

Moreover, I do not understand Referee's comment that "*... the finding of gapless quasiparticle excitations proximate to such a Luttinger surface is in direct conflict with the outcome obtained from the analysis of Heath et al. mentioned above. Indeed, Heath et al. appear to rule out such a possibility for a pseudogapped Luttinger surface*". As far as I can understand, Heath and Bedell do not exclude that a Luttinger surface, which carries a topological index like a Fermi surface, could also sustain gapless quasiparticle excitations. As a matter of fact, I am aware of a model where this result can be rigorously demonstrated, which is the impurity model I studied with Lorenzo De Leo in Phys. Rev. B **69**, 245114 (2004), equivalent to the two impurity model originally studied by Jones and Varma, Phys. Rev. Lett. **58**, 843 (1987). This model has an unscreened phase in which the impurity is pseudogapped because of a singular self-energy, and yet has a well defined gapless quasiparticle, in Nozières's local Fermi liquid sense, contributing, e.g., to a linear specific heat. Evidently, in an impurity model there is not a meaningful definition of the above mentioned topological index, nonetheless one can still define a Luttinger-Ward functional and correspondingly derive Luttinger's theorem. Therefore, since I really do not understand Referee's comment, I am equally unable to answer her/him. I apologise for that.

2. I thank the Referee for the Comment 2. that made me realise an aspect that was ill explained in the original version. In fact, my assumption $\Gamma(\epsilon \rightarrow 0, \mathbf{k}) \sim \epsilon^2$, now in Eq. (3), is closely related to another assumption, namely that $\Xi(\epsilon, \mathbf{k})$ defined in Eq. (5) of the revised version, has a regular Taylor expansion, at least to leading order, close to $\epsilon = 0$ and what I dub as Luttinger-Fermi momentum \mathbf{k}_{FL} . On the contrary, $\Xi(\epsilon, \mathbf{k})$ is non-analytic at $\epsilon = 0$ and $\mathbf{k} = \mathbf{k}_{FL}$ in non-Fermi liquids, like Luttinger liquids and marginal Fermi liquids.

In the revised version I better clarify the meaning of those assumptions, and mention explicitly the difference with Luttinger liquids and marginal Fermi liquids.

Another aspect that was probably unclear is that the existence of quasiparticles near a Luttinger surface derives just by those assumptions, and does not rely on the validity of Luttinger's theorem. I mentioned Luttinger's theorem just to clarify the meaning of the quasiparticle 'Fermi' volume with respect to the number of particles in the case of a Luttinger surface. To avoid any misunderstanding, in the revised version I first show that my assumptions about $\Gamma(\epsilon, \mathbf{k})$ and $\Xi(\epsilon, \mathbf{k})$ lead to the existence of quasiparticles at a Luttinger surface, and next I show how Luttinger's theorem looks in that case.

Moreover, I have changed the sentence mentioned by the Referee, which now reads "This result expands the class of interacting electron systems, like, e.g., Luttinger liquids, which are predicted to display conventional Landau-Fermi liquid behaviour, despite very different and anomalous single-particle properties.", thus removing the "substantially" and mentioning Luttinger liquids, which are well known to display the same long-wavelength low-energy linear response as Fermi liquids.

I hope that the above reply and the revised version satisfy Referee's requests, whom I thank again for her/his help in improving the manuscript.

Reviewers' Comments:

Reviewer #1:

Remarks to the Author:

After reading the revised paper and the responses to the referees (mine in particular) this is a significantly improved paper. His changes have made this more clear and easier to read for a more general audience. This paper satisfies some of the most important guidelines listed above: noteworthy, significant, sound methodology, and reproducible. In my original referee report I had two significant concerns that have been clearly addressed in this revised version. Given the revisions and the deeper insight of the paper; I recommend this paper for publication in Nature Communications.

Reviewer #2:

Remarks to the Author:

The authors carefully made revises and well addressed my concernings.

Reviewer #3:

Remarks to the Author:

I am satisfied with the responses offered by the author, as well as the changes made to the manuscript. The manuscript can now be accepted for publication.